# A Bibliometric Evaluation of the Use of Biomimicry as a Nature-Compatible Design Approach in Landscape Architecture Within the Context of Sustainability and Ecology

**DOI:** 10.3390/biomimetics10090559

**Published:** 2025-08-22

**Authors:** Rayan Ali, Deryanur Dinçer

**Affiliations:** Department of Landscape Architecture, Recep Tayyip Erdogan University, Rize 53020, Türkiye; rayan_ali98@hotmail.com

**Keywords:** nature, imitation, landscape, emulation, biomimetics

## Abstract

**Background:** The growing environmental crisis, driven by population increases and rapid urban development, has amplified the need for sustainable and ecological design approaches. Biomimicry, drawing inspiration from nature’s forms, processes, and systems, offers promising solutions in this context. Particularly in landscape architecture, biomimicry supports the integration of esthetics with ecological responsibility. **Methods:** This study presents a bibliometric analysis using the Scopus database to quantitatively assess the relationship between biomimicry and sustainable/ecological design within landscape architecture. A stepwise search strategy was applied, and the Biblioshiny tool within the version 4.2.1 of Bibliometrix package in RStudio 2024.04.1+748 software was used for data analysis and visualization. **Results:** A total of 1634 documents were identified under the keyword “biomimicry,” among which 210 addressed sustainability and/or ecological design. However, only three studies explicitly connected biomimicry, sustainable/ecological principles, and landscape architecture. Keyword trends, publication years, and country-level contributions were also examined. **Conclusions:** The findings highlight a substantial gap in the literature on the integration of biomimicry within sustainable landscape architecture. This underscores the need for further interdisciplinary research and practice that incorporates biomimetic principles to promote ecological innovation in landscape design.

## 1. Introduction

In recent years, as calls for action against the climate crisis, one of the world’s most pressing issues, have increased day by day, negative changes in the global climate have continued unabated. Although these calls have not resulted in definitive solutions, they have succeeded in creating global awareness. Nevertheless, the steps taken under so-called green policies to address the pace of climate change have proven insufficient and have failed to slow down this change [1,2,3]. Since the Industrial Revolution, the burning of fossil fuels for human activities has been the primary cause of the increase in greenhouse gas levels that has led to global warming. Since the 1950s, the current warming trend, resulting from human activities, has been progressing at an unprecedented rate [4]. We encounter numerous environmental sustainability problems both in production and consumption. According to the balance that exists in nature, it is evident that there is no ecosystem focused solely on production or solely on consumption. Although human existence that involves living conditions based on both production and consumption, humanity is continuing to accelerate the steps to destroy the Earth by consuming its resources in a short period, despite their long-lasting nature [5].

The main source of problems is that, although nature presents what is suitable, harmonious, functional, and permanent, humans have neither imitated nor taken inspiration from it [6,7]. The Earth, with its many complexities, has not been considered a place for non-human beings. There remains insufficient awareness that the Earth’s resources are shared with non-human species and materials and are limited [8]. However, since their existence, humans have felt a sense of belonging, affinity, and emotional attachment to nature and other living organisms [9,10]. Humans are dependent on nature and other beings for their survival. Throughout their existence, they have turned to nature to seek answers to problems. Thanks to the capabilities that distinguish humans from other living beings—reasoning, thinking, and analyzing—they have modeled natural events, other living beings, and inanimate objects in many areas [11].

As a thinking organism, humans inherently learn by discovering, through trial and error, by thinking and developing, and by imitating the sounds and behaviors they observe [12]. According to Aristotle, humans learn by imitating nature and other people [13,14]. Naturalist philosopher Rousseau also viewed nature as a true educator, guide, and instructor for humans [15,16]. In contemporary culture, considering nature’s influence on human psychology, the prevailing idea is that nature is a healer. Numerous studies demonstrate the role of nature in treating contemporary psychological illnesses such as depression, stress, anxiety, and attention disorders [17,18,19,20]. In contrast, many professional disciplines such as engineering, architecture, and design tend to be human-centered, serving almost exclusively human needs. As a result of this anthropocentric approach, principles like human progress in design contribute to the consumption of non-human elements, the ongoing extinction of other species, and the ecological reality of climate change [8].

Wakkary states that design must be responsible for the life it constructs, asserting that it shapes both our world and our understanding of what it means to be human [21]. Sustainable development, defined in the 1980s to address environmental problems arising from industrial activities and to create a society that meets human needs without harming the integrity of natural systems, brought sustainability into the realm of design [22,23,24,25]. Consequently, the necessity of prioritizing sustainable and ecological criteria in design became increasingly emphasized [25]. Thus, through the practice of emulating nature, efforts to find sustainable solution methods offered by nature were initiated [5].

### 1.1. Definition of Biomimicry

Today, scientific advancements allow us to clearly observe how life is sustained, the forms it takes, and how it functions within ecosystems, from the largest to the smallest components [6,7]. Biomimicry is the discipline that reads nature and studies the functions occurring within it. It identifies natural forms, processes, and ecosystems, then imitates them to create sustainable designs [26]. In other words, biomimicry is a discovery tool encompassing both design and ecological disciplines, aiming to solve problems by emulating nature [6,27]. Ecological and sustainable design is an extension of biomimicry [28]. Nature, like an open encyclopedia, contains countless organisms that have passed the test of experience, playing roles such as engineers, chemists, and physicists, as well as economic communities and ecosystems living in harmony and cooperation [26]. According to Pawlyn, Buckminster Fuller’s goal of making the Earth work for 100% of humanity without environmental harm or disadvantaging anyone, as quickly as possible through cooperation, can be achieved through biomimicry, which views nature as a genius [29].

In the design context, biomimicry involves carefully and consciously observing nature’s forms, functional structures, material compositions, and processes to develop design proposals that meet human needs and facilitate life [28]. The solutions offered by nature have always guided designers. New designs can be created by drawing inspiration from nature’s perfect designs [14,30]. Nature serves as a source of inspiration for designers to find solutions to problems and transform them into forms. It also offers an infinite data source for designers seeking formal exploration, in terms of visual form, harmony, and order. As designers understand the relationships in nature, their repertoire of forms becomes enriched [14].

Benyus associates the biomimicry revolution with sustainability and emphasizes that, unlike previous revolutions, it is not about stealing nature’s secrets, ruling over it, or domesticating it [6,7]. Instead, it invites humility, encouraging humans to approach nature as part of it and to learn the sustainable and ecological secrets that enable harmonious living through biomimicry [6,26]. Thus, in a broader definition, biomimicry is the application of studying nature’s forms, processes, and ecosystems, learning strategies found in nature, and drawing inspiration to create sustainable designs, human-centered design activities, and solutions for complex human problems [6,7]. In this way, biomimicry has named a new discipline that imitates nature’s designs and processes to create a healthier and more sustainable planet [31]. In short, “Biomimicry is an esthetic practice concerned with the practical dimensions of sustainability” [32].

### 1.2. Concepts Commonly Confused with Biomimicry

Before the term biomimicry, the terms biomimetics and bionics were used [29]. Bionics, biomimetics, and biomimicry are generally presented as three bio-inspired disciplines that are often not clearly distinguished from one another. Most researchers agree that these three approaches share similarities, particularly in learning from nature with a focus on innovation and technology. However, a key difference of biomimicry compared to other bio-inspired disciplines is that one of its main goals is the preservation of life and nature, thus closely relating it to environmental sustainability [33].

### 1.3. Levels of Biomimicry

There are three levels of biomimicry: form, process, and ecosystem [34,35,36]. The aim of these levels is not to replicate a form, process, or ecosystem exactly, but to use the design principles of a natural form, process, or ecosystem as a trigger for idea generation. Designers aiming to create more sustainable designs through biomimicry should attempt to imitate biological strategies on all three levels—form, process, and ecosystem. According to researchers, this multi-level approach is the most effective way to achieve successful solutions in terms of sustainable performance [37]. Indeed, Zari defined biomimicry as a tool to enhance the sustainability of human-made product designs, materials, and the built environment [35].

### 1.4. The Relationship of Landscape Architecture with Nature, Ecology, and Sustainability

According to Murphy, landscape is a complex term, referring to a designed environment, a natural scene, or an ecological system that organizes energy and matter [38]. Landscape architecture, a field encompassing urban and regional planning, landscape planning, and design, is directly related to scientific disciplines such as geography, geology, hydrology, soil science, and ecology [39]. According to Adedeji, in landscape design, the influence of basic components of the natural world climate, vegetation, water, topography, relief, drainage patterns, and soil should not be overlooked [40]. Although user needs form the foundation of the design process, the complex interaction of climate, geology, vegetation, wildlife, and other elements of the natural environment must be considered in landscape design. This approach is necessary to ensure that landscape design is in harmony with natural processes and sustainably meets user needs. When introducing the roles of landscape architects, IFLA emphasized ecological sustainability. According to this definition, landscape architects plan, design, and manage natural and built environments while considering values such as ecological sustainability, esthetic, scientific principles, and social justice [41]. They also address factors such as climate change, ecosystem stability, socio-economic developments, and community health. In summary, landscapes are formed by natural systems but are also shaped by history and culture [40].

Landscape architecture and biomimicry are both related to design and nature and according to literature research, sustainable and ecological design seems to be a deeper common point of both areas.

The aim of this study is to raise awareness about biomimicry and its relationship with landscape architecture within the context of sustainable and ecological design and quantitatively determine the presence of biomimicry, its relationship with sustainable/ecological design, and its application within the field of landscape architecture as reflected in the literature. By conducting a bibliometric analysis, the study seeks to reveal the link between biomimicry and both esthetic and ecological-sustainable design approaches, particularly in landscape architecture, and to provide insights that may guide future researchers and support their evaluation processes. In this context, stepwise assessments were conducted using the Scopus database. In the first stage, a bibliometric analysis was carried out focusing on biomimicry (Figure 1); in the second stage, biomimicry in the context of sustainable and/or ecological design was analyzed (see Figure 1); and finally, the current state of the use of biomimicry in sustainable and/or ecological design within landscape architecture was examined (see Figure 1).

## 2. Method

Within the scope of the study’s objective, step-by-step evaluations were conducted in a manner proceeding from the general to the specific in order to quantitatively reveal the use of biomimicry in the context of sustainable and/or ecological design within the field of landscape architecture through academic studies. In line with the aim of this study, the following aspects were examined through academic publications available in the database using a bibliometric analysis method:The current status of academic studies conducted on the subject of biomimicry;The current status of academic studies addressing biomimicry together with sustainable and/or ecological design;The current status of academic studies addressing biomimicry together with sustainable and/or ecological design within landscape architecture.

The workflow of the study was designed based on the standard science mapping workflow proposed by Zupic [42], and was inspired by the studies of Aria et al. [43] and Varshabi et al. [44], forming a three-stage process (Figure 1).

**Figure 1 biomimetics-10-00559-f001:**
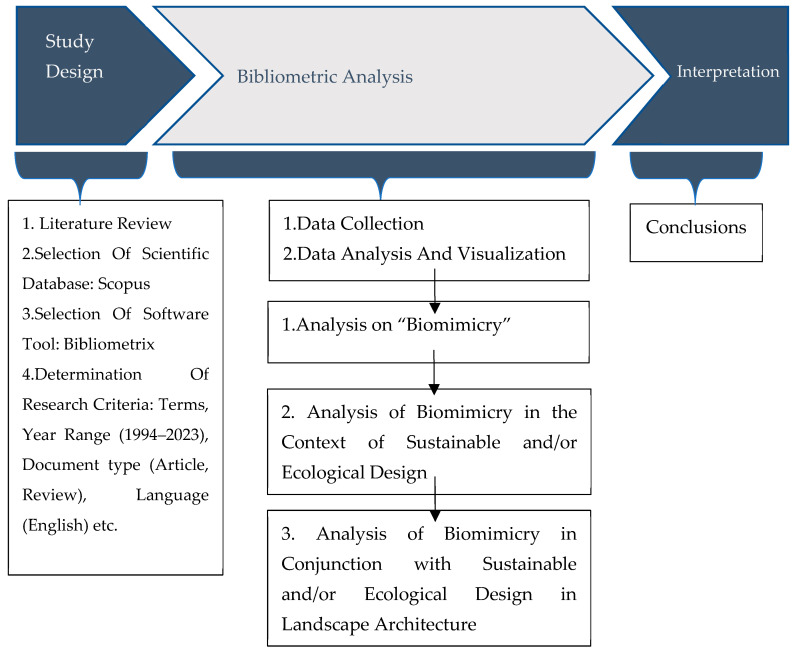
Workflow of the method applied in the study.

### 2.1. Bibliometric Survey

The bibliometric analysis method provides qualitative and quantitative evaluations of academic publications, authors, and institutions using statistical techniques. Various criteria such as citations, impact factors, and time interval data are employed in these methods [45]. Properly planned and implemented bibliometric studies provide a solid foundation for advancing a scientific field in a more accurate and meaningful way. Through bibliometric studies, researchers can gain an overview of a particular field, identify knowledge gaps, generate new ideas for research, and correctly position the intended contributions to the field [46].

Biblioshiny is a web-based application integrated into the Bibliometrix package. This tool facilitates the use of Bibliometrix by non-coders and is particularly user-friendly for users without programming experience (URL-2). Bibliometric analyses are conducted by extracting data from appropriate databases. The reliability and validity of the results obtained through bibliometric analysis depend directly on the scope, currency, and indexing policy of the selected database [47]. Scopus and the Web of Science Core Collection are two prominent, multidisciplinary bibliographic databases frequently utilized in academic research. Although both are widely cited in scholarly publications, Scopus—despite being a relatively newer entrant—has increasingly challenged the dominant position long held by Web of Science [48]. Among scholars in library and information science, it is well established that both the established Web of Science and the rapidly growing Scopus play a central role in bibliometric analyses [49,50,51].

According to Elsevier (2023), Scopus is one of the largest abstract and citation databases of peer-reviewed literature, and covers over 25,000 active titles from more than 5000 international publishers, encompassing a broad spectrum of disciplines including science, technology, medicine, social sciences, and arts and humanities [52]. it is open-access and possesses extensive data sources across a wide range of subjects, allowing for advanced search capabilities [53]. Compared to other databases such as the Web of Science, Scopus offers broader journal coverage, making it particularly valuable for interdisciplinary and international research evaluations [54,55].

In the data collection phase of this study, the Scopus database was used with the aid of appropriate terms.

### 2.2. Preliminary Survey of Database

In this study, the term “biomimicry” was selected as the primary word in the Scopus database search. The examination of documents was carried out in May 2024. A search using the term “biomimicry” without any limitations produced 16,216 results in Scopus, spanning from 1982 to 2024, including articles, reviews, conference papers, and book chapters. However, the research aims to visualize the knowledge structure over a micro field. Therefore, the search underwent several filtering processes. Furthermore, considering that databases are regularly updated, all related data were collected on a single day in May 2024.

### 2.3. Selection and Customization of Terms

In the first stage, a search was conducted in the Scopus database using the term “biomimicry” within the fields of Article Title, Abstract, and Keywords. As a result of this search, 2718 documents published between 1982 and 2024 were retrieved. The obtained data provided a broad initial sample on the subject. This sample was then subjected to a filtering process: only documents categorized as Articles and Reviews were included under Document Type; only sources classified as Journals were retained under Source Type; and only fully published documents (excluding Articles in Press) were considered in terms of publication status. Regarding language, only documents published in English were included. No geographical restrictions were applied. At the end of this stage, a final sample consisting of 1634 academic publications published between 1994 and 2023 was obtained.

The second stage involved refining the sample obtained in the previous step. The Scopus search engine supports Boolean syntax, which allows users to combine terms using operators such as AND, NOT, and OR to produce more specific results [56]. The operator AND is used to narrow down results by requiring all combined terms, whereas OR is used to broaden results by including any of the specified terms. This method is particularly useful when the research topic includes terms with multiple meanings, as it helps to focus the search and combine relevant concepts to identify the exact subject of interest [57]. At this stage, more targeted searches were conducted using the Boolean logic supported by the Scopus database. The term “biomimicry” was combined with other terms relevant to the research scope—such as “design,” “sustainability,” “ecology,” and “landscape architecture”—as well as their equivalent terms listed in Table 1, using AND and OR operators.

### 2.4. Data Collection, Analysis, and Visualization

The data collection phase included creating a csv file from the filtered datasets obtained through the two-stage search in the Scopus database, which were planned for bibliometric analysis.

At this stage, to conduct a descriptive bibliometric analysis, 4.2.1. Bibliometrix package codes in the 2024.04.1+784 version of RStudio software environment were used. The CSV-format dataset containing publication title, publication year, journal, author, keyword, abstract, and citation data was uploaded into the Biblioshiny program for analysis and visualization in line with the research questions.

## 3. Findings

### 3.1. Findings Related to the Number of Documents in the Scopus Database

Table 1 presents the results of a series of refined searches conducted in the Scopus database using the term “biomimicry,” combined with various other term groups to align with the objectives of the study. The distribution of the refined search areas is illustrated using arrows in different colors.

As shown in Table 1, an initial broad search using the term “biomimicry” yielded 1634 documents. The search was then progressively refined through the following steps:First, combining “biomimicry” with “design” reduced the number of documents to 702.Second, to examine the relationship between biomimicry and landscape architecture, a new search was conducted by combining “biomimicry” with “landscape architecture” and related terms, resulting in four documents.Third, the 702 “biomimicry+design” related documents were further refined in three sub-steps:

Searching with “design + biomimicry + landscape architecture” and related terms yielded three documents;

Searching with “design + biomimicry + sustainability” yielded 155 documents;

Searching with “design + biomimicry + ecology” yielded 104 documents.

Fourth, combining “design + biomimicry + either sustainability or ecology” yielded 210 documents.Fifth, combining “design + biomimicry + sustainability and ecology” simultaneously yielded 49 documents.Sixth, adding “landscape architecture” to the fourth-stage keyword combination yielded three documents.Seventh, searching with “design + biomimicry + sustainability + landscape architecture” and related terms yielded one document.Eighth, using “design + biomimicry + ecology + landscape architecture” and related terms yielded two documents.Ninth, the most specific combination—“design + biomimicry + ecology + sustainability + landscape architecture” and related terms—resulted in no documents being found (Table 1).

In Figure 2, the fields have been transformed into a boundary map based on the search results provided in Table 1.

According to the general findings obtained from the analysis related to biomimicry, out of the 1634 documents published between 1994 and 2023, 1262 are articles and 372 are reviews. These documents have been published across a total of 827 journals. The total number of cited references in these documents is 104,583. Additionally, the total number of author keywords associated with the documents is 4856. The total number of authors is 5856, with 164 of the documents being single-authored.

Based on the analysis involving biomimicry, design, sustainability, and ecology collectively, it was found that no document data related to the subject area existed in the Scopus database between 1994 and 2006. Therefore, the period from 2006 to 2023 was included in the evaluation. Among the 210 documents obtained, 171 were articles and 39 were reviews. Of these, 48 documents were written by a single author, and the total number of authors for these single-authored papers is 38. The total number of author keywords across the 210 documents was 757. The total number of authors contributing to these documents was 648. The total number of references used in the 210 documents was 11,808. Additionally, these documents were published across 144 different journals.

### 3.2. Annual Publications Trend

Annual scientific production based on the data of 1634 documents related to the term “biomimicry” and 210 documents that include biomimicry, design, sustainability and/or ecology collectively is shown in Figure 3a,b.

According to Figure 3a, within the limitations of the study, it was found that the production of documents related to the subject of biomimicry began slowly in 1994 and remained quite limited and slow until 2004. However, from 2004/2005 onward until 2023, a significant and rapid increase in development has been observed.

Based on the analysis involving biomimicry, design, sustainability and/or ecology collectively, it was observed that scientific production began in 2006 and continued with a steady increase between 2006 and 2023 (Figure 3b).

### 3.3. Analysis of the Relationship Between Authors (AU), Author Countries (AU_CO), and Journals (SO) Based on Document Data

Three-field plots showing the connections between Authors (AU), Author Countries (AU_CO), and Journals (SO) are presented in Figure 4a,b.

Upon examining Figure 4a,b, it is observed that *Biomimetics* is the most frequently preferred journal in both analyses. According to the analysis using the keyword biomimicry, Chinese authors—who produced the highest number of documents on the subject—did not prefer the journal *Biomimetics* the most; instead, they most frequently chose *ACS Applied Materials and Interfaces, Advanced Functional Materials*, and *ACS Nano*, respectively. It was found that American authors tended to prefer all journals at nearly equal rates (Figure 4a).

In the analysis involving biomimicry, design, sustainability, and ecological design, it was observed that the most productive authors, from the United States, most frequently preferred the journal *WIT Transactions*, followed by *Biomimetics* and *Acta Biomaterialia*. Authors from France, Australia, New Zealand, and the Netherlands were found to prefer *Biomimetics* the most (Figure 4b).

### 3.4. Geographical Regions Analysis

According to the analysis of 1634 documents based on the keyword biomimicry, the ranking of countries by scientific productivity—based on the nationalities of all contributing authors—places the United States first, followed by China in second, the United Kingdom in third, Italy in fourth, and Australia in fifth (Figure 5a).

According to the analysis of 210 documents involving biomimicry, design, sustainability and/or ecology, when examining the scientific productivity of countries over time from 2006 to 2023, it is seen that countries other than the United States progressed at nearly the same level and at a slower pace. In contrast, the contribution of authors affiliated with the United States in this subject area demonstrated a much faster and significantly higher rate of development compared to other countries (Figure 5b).

The spread of the research work and the geographical distribution of publications are shown in the map in Figure 6a,b. The color intensity is proportional to the scientific productivity of the countries. Figure 6 shows that the US colored with dark blue is the most productive country.

### 3.5. Author Keywords, Title, and Subject Area Analysis

While identifying the most frequently used words and their frequencies in author keywords and titles, the terms biomimicry and biomimetics were excluded from the analysis. This is because these terms were selected as primary search terms for the study and were considered equivalent. Their exclusion aimed to ensure that the visualizations would be more specific and informative. Table 2 shows the words most used in author keywords and titles of 1634 document’s set.

In documents related to the field of biomimicry, among the author keywords, Bioinspiration (93 occurrences) ranks first, Sustainability (69 occurrences) ranks third, Design (30 occurrences) ranks sixth, and Architecture (25 occurrences) ranks tenth among the ten most frequently used terms (see Table 2).

Regarding document titles, Design ranks first with 208 occurrences, followed by Bioinspiration in second place with 135 occurrences and Sustainability in seventh place with 68 occurrences (see Table 2).

Figure 7a,b present word clouds of the most frequently used words in author keywords and titles of the 1634 documents, respectively,

Figure 8a shows the annual trend topics of the documents based on the author keywords from 1634 documents. In Figure 8a, the lines represent the temporal placement of the topics on the timeline, while the nodes indicate the frequency of the keywords. Additionally, the extension of the lines suggests that the topics have remained relevant over time [58].

In Figure 8b, the keyword group Biomimicry–Sustainability–Ecology–Design was removed to better highlight the main keywords of the collection and allow for a more specific visualization. Furthermore, the terms within the groups Biomimicry–Biomimetics–Biomimetic, Bio-inspired–Bio-inspiration, and Bioinspired design–Biologically inspired design were treated as synonyms. Based on this grouping, the trending research topics over the years were identified from the 210 documents using author keywords.

According to the author keywords from the 1634 documents, the most frequently used terms related to biomimicry in 2023 were “bioink” and “porosity”. Additionally, “soft robotics” and “simulation” emerged as trending topics in 2022 and remained relevant through 2023. Starting from 2016, “3D printing” maintained its relevance until 2022, peaking in 2021. Similarly, “design” became prominent from 2014, peaking in 2020, and “biologically inspired design” appeared from 2014 with a peak in 2018, continuing as a trend until 2022. The keywords “sustainability” (since 2014), “architecture” (since 2016), and “bioinspiration” (since 2015) were also observed to remain current up to 2021. Based on keyword frequency, the terms with the highest frequency by year were “bioinspiration” in 2018, “sustainability” in 2019, and “tissue engineering”, also in 2019.

In Figure 8b, the most trending topics in 2019 were “sustainable design”, “innovation”, and “architecture”, followed by “regenerative design”, “built environment”, and “ecosystem services” in 2020. In 2021, “urban design” became the dominant trend. Among these, “urban design” (2021), “ecosystem services” (2020), and “sustainable design” (2019) remained relevant up to 2022.

### 3.6. Network Analysis Using Clustering

The thickness of the circles is proportional to the frequency of use, while the thickness of the connecting lines indicates the frequency of co-occurrence.

#### 3.6.1. Co-Occurrence Network Analysis of Author Keywords

According to the co-occurrence network of author keywords, the term “biomimicry” is frequently used alongside many other terms. Additionally, “bioinspiration”, “biologically inspired design”, and “sustainability” are often used together (Figure 9).

#### 3.6.2. Author Co-Citation Network Analysis

Figure 10a shows the author co-citation network based on the data from 1634 documents, which is clustered into three main groups. This indicates that the authors tend to concentrate on three primary thematic areas. In Figure 10b, based on the data from 210 documents, the co-citation network is clustered into four main groups, suggesting that the authors focus on four distinct thematic areas.

The most frequently co-cited author pairs include the following: Benyus J–Vincent J, Benyus J–Zari P, Benyus J–McDonough W, Benyus J–Baumeister D, Zari P–Reed B, and Zari P–Hayes S.

#### 3.6.3. Document Co-Citation Network Analysis

According to Figure 11a, based on the data from 1634 documents, the most locally cited references are Benyus (1997) and Pawlyn (2011), making them the most frequently co-cited documents. In Figure 11b, the most commonly co-cited document pairs are Benyus (1997)–Pawlyn (2011), Pedersen Zari (2018)–Baumeister (2013), and Benyus (1997)–Helms (2009).

#### 3.6.4. International Collaboration Network Analysis

Based on the data from 1634 documents, the most intensive international collaborations in this field are observed between the United States and China, the United States and the United Kingdom, and the United States and Italy (Figure 12a). According to the data from 210 documents, the strongest collaboration is found between the United States and Australia (Figure 12b).

### 3.7. Findings Related to the Literature on Sustainable and/or Ecological Design in Landscape Architecture in the Context of Biomimicry

As a result of the search conducted within the limitations of this study, a total of three documents were identified that are related to biomimicry, design, and landscape architecture. Among these, two documents were found to be associated only with ecological design and its equivalents, while one was linked solely to sustainable design and its equivalents. However, according to the refined search results, no documents were found that simultaneously address biomimicry, design, landscape architecture, sustainability, and ecology (Table 1). The documents obtained from Scopus are as follows:*The Avifauna-Based Biophysical Index (ABI) approach for assessing and planning ecological landscaping in tropical cities* [59].

According to this document most well-regarded ecological landscape initiatives have been concentrated in developed countries within temperate regions, which may result in systemic approaches falling short in capturing and reflecting natural processes in tropical areas. In his study, the author addressed this gap by introducing and testing a new approach that utilizes birds as indicators to guide ecological landscaping in tropical cities. Specifically, he applied a set of avifaunal metrics to assess the biophysical complexity of six representative strata in and around the city of Ipoh, Malaysia [59].

*Environmental tendencies in modular green installations* [60].

According to this document living green walls encompass two main approaches: ecological and artistic. In their study, the authors aimed to enhance previous experiences by analyzing methods that integrate both aspects into a functional, ecological, and simultaneously esthetic composition. The authors proposed a new concept of green installations composed of interactive modular systems that merge into a living, dynamic, mobile, and responsive structure inspired by nature and guided by biomimicry as a core design principle. They argue that this new concept responds to and is influenced by both external natural stimuli and the human factor. Furthermore, they emphasize that the development of this project—integrating architecture, art, interior and landscape design, botany, geometry, mechanical and electrical engineering—relies heavily on interdisciplinarity as a key element [60].

*Development of sustainable landscape design guidelines for a green business park using virtual reality* [61].

In this document the authors emphasized that sustainable landscapes are a continuous necessity in Green Business Parks, which aim to ensure productive and healthy work environments that are attractive to employees. They highlighted the significant role of using Virtual Reality (VR) techniques to gather feedback from users and experts, as a means to accurately predict the performance of a given space. Moreover, the primary objective of their study was to develop guidelines for sustainable landscape design in future Green Business Parks. In their research, three different landscape design models—formal, xeriscape, and biomimicry-based—were developed as 2D and 3D models using AutoCAD and SketchUp. These alternatives were subsequently evaluated by experts and users through the use of VR headsets and mobile devices [61].

Although these studies produced positive outcomes, they did not change the fact that, within the scope of this research, there remains an insufficiency of research in the field of Landscape Architecture where biomimicry is comprehensively applied for sustainable/ecological design.

## 4. Results and Discussion

In this study, a bibliometric analysis was conducted to examine the presence of biomimicry in the academic literature and its connection with sustainable and ecological design approaches in the field of landscape architecture. Notably, Kshirsagar (2021) emphasized the lack of a systematic bibliographic investigation in the field of biomimicry [48].

Human beings are inherently dependent on nature and other living beings to sustain life. Through the unique ability to think, reason, and analyze, humans have modeled many natural events and entities. Biomimicry, as a nature-inspired concept, is frequently used interchangeably with bionics and biomimetics, constituting the three primary disciplines of bio-inspiration. However, a defining feature of biomimicry is its specific aim of preserving life and nature, thereby associating it directly with environmental sustainability [33]. In this context, Benyus defines biomimicry as the integration of nature’s ecological and sustainable principles into human life. Therefore, this study adopted keywords such as “biomimicry,” “design,” “sustainability,” “ecology,” and “landscape architecture,” along with their equivalents, to ensure alignment with the research purpose [6].

The Scopus database search using these keywords revealed 1634 documents containing the keyword “biomimicry.” When combined with the term “design,” the number reduced to 702. Further refinement by combining “biomimicry,” “design,” “sustainability,” and “ecology” yielded a sample of 210 documents.

Scientific output on biomimicry between 1994 and 2023 began to increase significantly after 2004 and continued to grow through 2023. This trend aligns with the escalating global concerns surrounding climate change and environmental degradation. Çepel (1988) noted that advanced technology, while improving human comfort, has also contributed to serious environmental issues [62]. Similarly, NASA (2024) emphasized that the current warming trend, primarily attributed to human activities, is proceeding at an unprecedented rate not observed for millennia [4]. This growing disruption of ecological balance has intensified interest in nature-inspired design solutions. Bhamra et al. (2021) also pointed out the rising importance of incorporating sustainable and ecological criteria into design practices [25]. The keyword trend analysis of 1634 documents showed that biomimicry and sustainability were prominent between 2014 and 2021, while “design” remained relevant between 2014 and 2022, and “architecture” between 2016 and 2021. In the subset of 210 documents, trending topics identified through author keywords and titles included sustainable design, innovation, and architecture in 2019; regenerative design, the built environment, and ecosystem services in 2020; and urban design in 2021. Topics such as “urban design” (2021), “ecosystem services” (2020), and “sustainable design” (2019) remained relevant through 2022. The frequent appearance of “tissue engineering” and “engineering” among the top keywords and titles suggests that biomimicry is most commonly applied in engineering disciplines.

Scientific productivity by country, as measured by all contributing authors per publication, showed the United States ranking first, followed by China and the United Kingdom. Although the UK began publishing in this field earlier (1994), it was surpassed by the U.S. in 2001 and by China in 2014 [63]. Country collaboration analysis showed the most intense cooperation occurred between the U.S. and China, the U.S. and the UK, and the U.S. and Italy. In the subset of 210 documents, the strongest collaboration was observed between the U.S. and Australia. In both datasets, Biomimetics was the most frequently selected journal, followed by WIT Transactions on Ecology and the Environment and Bioinspired Biomimetics and Nanobiomaterials.

Regarding the primary focus of this study the presence of biomimicry in the context of sustainable and/or ecological design within landscape architecture only three relevant documents were found. Two of these addressed ecological design (or its equivalents), and one focused on sustainable design. No publication was found that integrated all five concepts simultaneously: biomimicry, design, landscape architecture, sustainability, and ecology. This finding indicates that the use of biomimicry in sustainable/ecological landscape design remains quite limited. While acknowledging the constraints of relying solely on the Scopus database and the specific filtering criteria applied, the author suggests considering other potential explanations. For instance, implementing biomimetic design strategies in landscape architecture may be limited by the complexity of ecosystems and practical challenges in the field.

Gebeshuber [64] emphasized that designers, engineers, and planners must possess at least a fundamental understanding of how ecosystems work. However, many design projects are led by professionals without ecological training, with limited resources for acquiring such knowledge, and under intense pressure to deliver results quickly and cost-effectively. Consequently, these conditions often result in superficial, form-based biomimicry implementations, as observed by Armstrong (2009), potentially lacking in sustainability performance [65]. Zari (2014) also stated that achieving adequate protection of ecosystem services is difficult due to the lack of strong regulations or market-based solutions [66]. As such, regenerative design often relies heavily on the goodwill of developers, planners, and designers. This reliance, coupled with the complexity and public-good nature of ecosystems, makes attaining effective ecological protection extremely challenging.

Zari (2012) recommended that designers intending to use ecosystem-based biomimicry should first identify an appropriate ecological focus specific to the site, ideally in consultation with local ecologists. Such processes would require broad interdisciplinary input and collaboration [67].

Despite these challenges, increasing the use of biomimetic strategies in the built environment can offer effective responses to intensifying climate-related problems, such as population growth, global warming, and urban heat island effects. Seçkin et al. (2011) defined a successful landscape design as one that considers both beneficial and harmful aspects of natural processes while protecting and enhancing the health of living organisms [68]. Araque et al. (2021) highlighted that traditional methods are no longer sufficient to address sustainability challenges, urging a paradigm shift in design thinking [69]. They advocated for adopting biomimetic strategies to solve problems as efficiently as nature does. Given the escalating environmental pressures, the encouragement of biomimicry-based design research is essential.

In summary, this study used bibliometric analysis to evaluate the current literature on the relationship between biomimicry and sustainable/ecological design within landscape architecture. It also aimed to guide future academic work by identifying key publications and mapping research trends. As Doğan (2019) noted, bibliometrics is a quantitative research method that measures and analyzes publications [70]. Similarly, Zupic (2015) emphasized that bibliometric methods can help understand both the intellectual structure and conceptual framework of a topic, thereby facilitating evaluation processes and guiding future research [42].

## Figures and Tables

**Figure 2 biomimetics-10-00559-f002:**
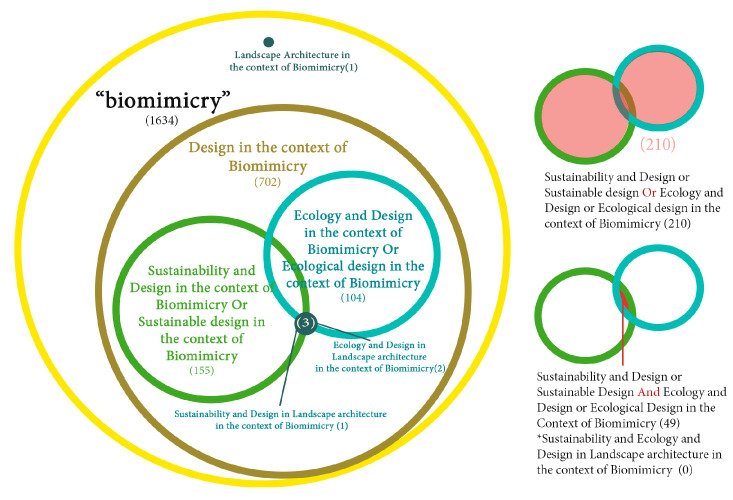
Boundary map of fields based on the number of documents in the Scopus database.

**Figure 3 biomimetics-10-00559-f003:**
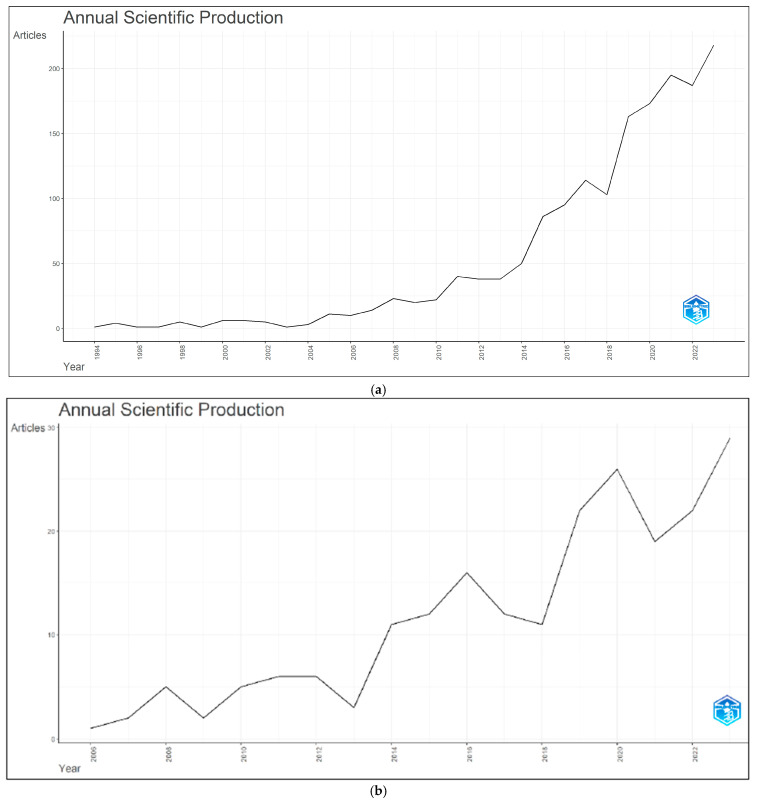
(**a**) Annual scientific production based on the data from 1634 documents obtained through analysis using the keyword biomimicry. (**b**) Annual scientific production based on the data from 210 documents in which biomimicry, design, sustainability and/or ecology are collectively addressed.

**Figure 4 biomimetics-10-00559-f004:**
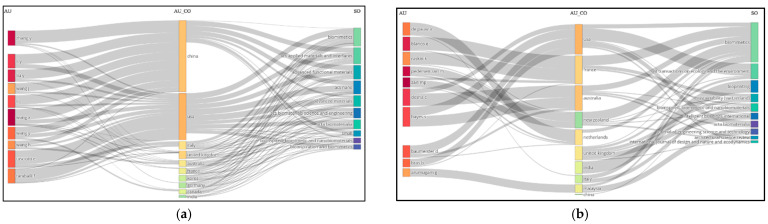
(**a**) Three-field plot based on the data from 1634 documents analyzed using the keyword biomimicry. (**b**) Three-field plot based on the data from 210 documents in which biomimicry, design, sustainability and/or ecology are collectively addressed.

**Figure 5 biomimetics-10-00559-f005:**
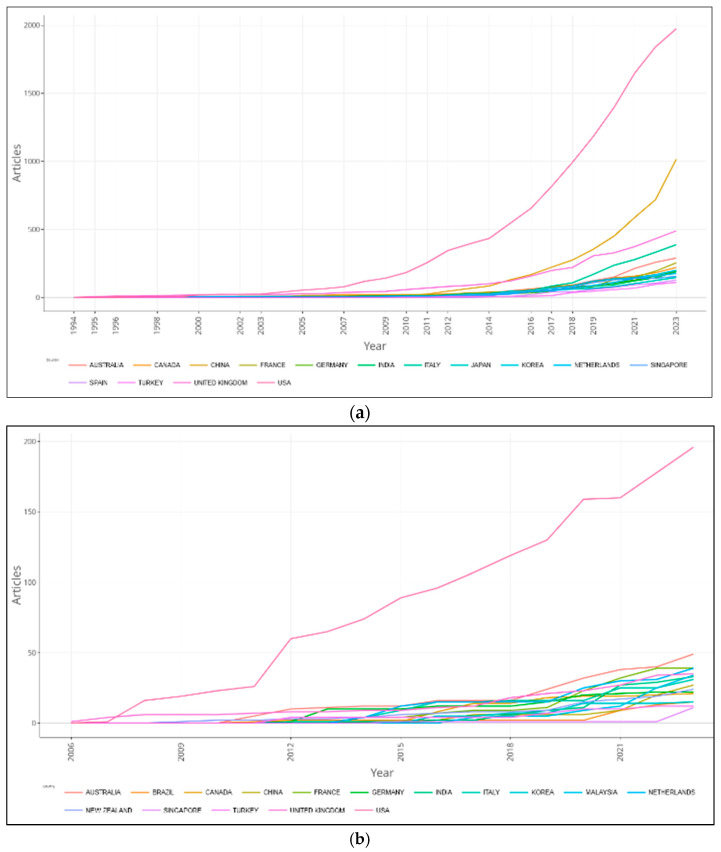
(**a**) Countries’ scientific productivity over time based on the data from 1634 documents analyzed using the keyword biomimicry. (**b**) Countries’ scientific productivity over time based on the data from 210 documents.

**Figure 6 biomimetics-10-00559-f006:**
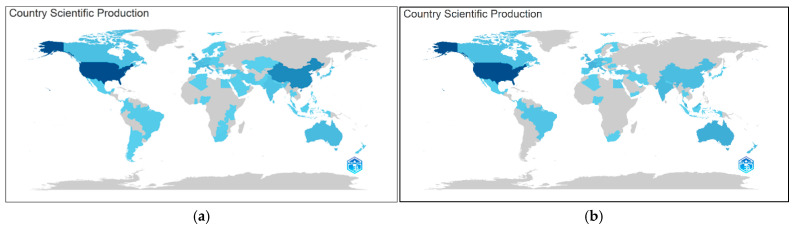
(**a**) Map of the countries’ scientific productivity over time based on the analysis of 1634 documents retrieved using the term “biomimicry”. (**b**) Map of the countries’ scientific productivity over time based on the analysis of 210 documents.

**Figure 7 biomimetics-10-00559-f007:**
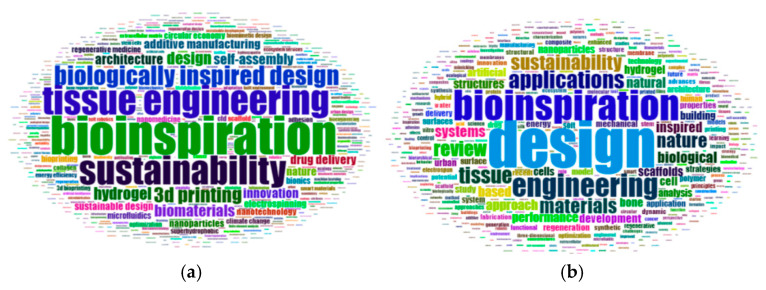
(**a**) Most frequently used author keywords associated with the keyword biomimicry. (**b**) Most frequently used word groups in document titles associated with the keyword biomimicry.

**Figure 8 biomimetics-10-00559-f008:**
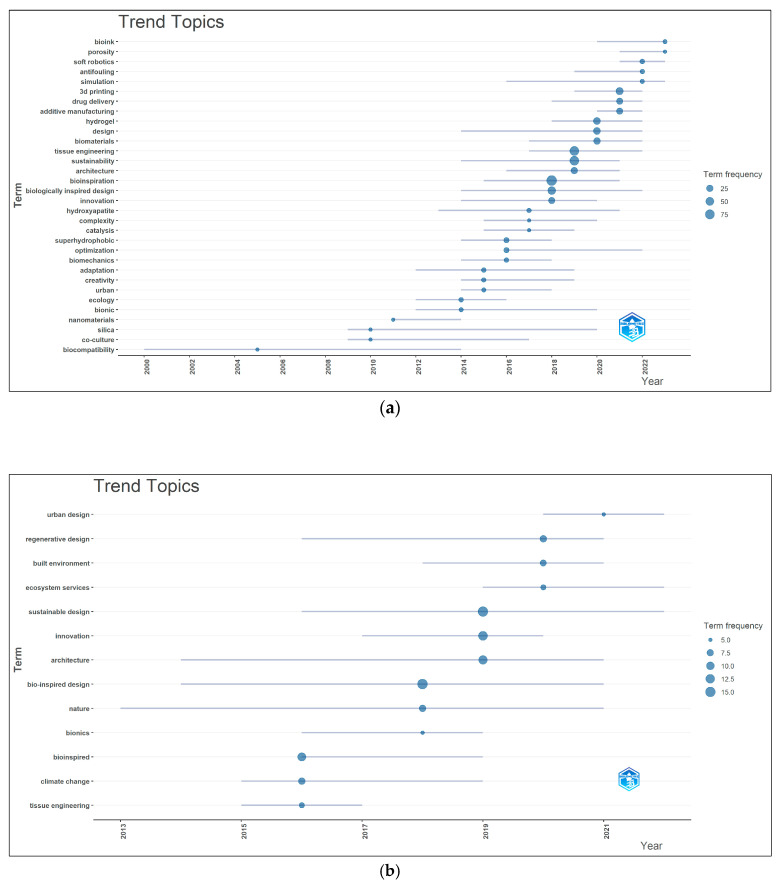
(**a**) Trending research topics and their frequencies over time based on author keywords from 1634 documents. (**b**) Trending research topics and their frequencies over time based on author keywords from 210 documents.

**Figure 9 biomimetics-10-00559-f009:**
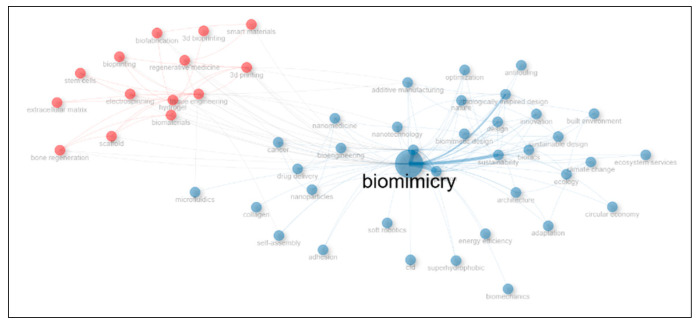
Co-occurrence network of author keywords based on the data of 1634 documents set.

**Figure 10 biomimetics-10-00559-f010:**
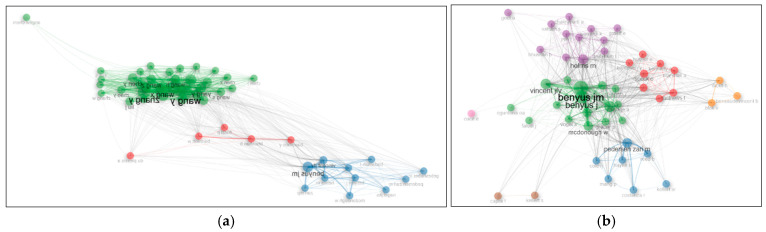
(**a**) Author co-citation network based on 1634 documents; (**b**) author co-citation network based on 210 documents.

**Figure 11 biomimetics-10-00559-f011:**
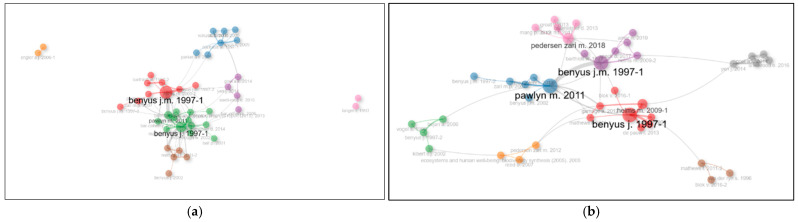
(**a**) Co-citation network of 1634 documents; (**b**) co-citation network of 210 documents.

**Figure 12 biomimetics-10-00559-f012:**
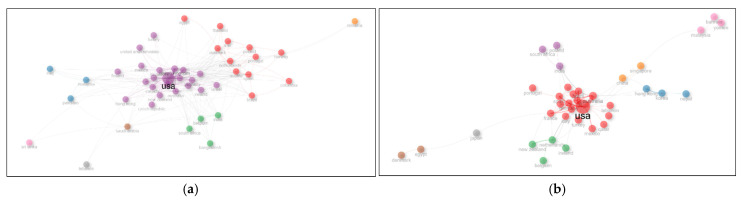
(**a**) International collaboration network based on 1634 documents; (**b**) international collaboration network based on 210 documents.

**Table 1 biomimetics-10-00559-t001:** Search results from the Scopus database regarding the refinement of the biomimicry search field using the terms Design, Sustainability, Ecology, Landscape Architecture, and their equivalent terms.

	Aimed Search Limitations	Terms	Number of Documents	
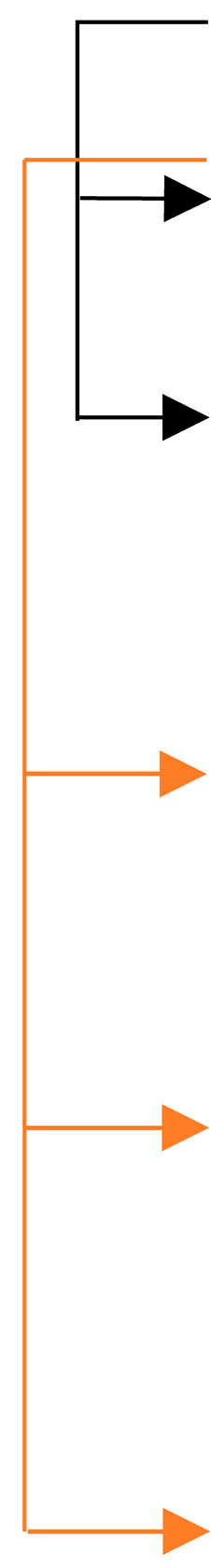	Biomimicry	“Biomimicry”	1634	
Design in the context of Biomimicry	“Biomimicry” AND “design”	702	
Landscape Architecture in the context of Biomimicry	“biomimicry” AND (“landscape architecture” OR “landscape design” OR “landscaping” OR “landscape planning”)	4	

(Design in the context of Biomimicry) And (Landscape Architeture)	(“biomimicry” AND “design”) AND (“landscape architecture” OR “landscape design” OR “landscaping” OR “landscape planning”)	3	
(Sustainability and Design in the context of Biomimicry) Or (Sustainable design in the context of biomimicry)	(“biomimicry” AND “design”) AND (“sustainability” OR “sustainable” OR “sustainable design”)	155	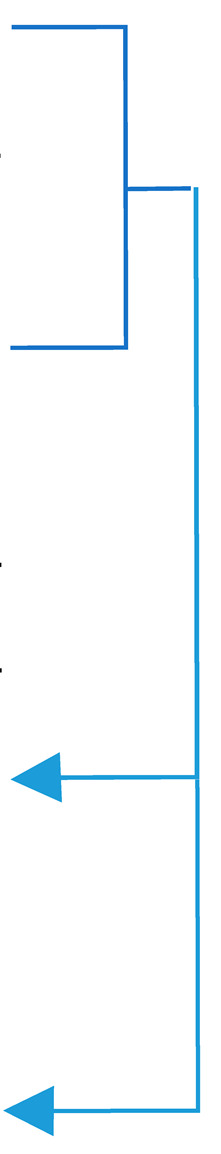
(Ecology an Design in the context of Biomimicry) or (ecological design in the context of Biomimicry)	(“biomimicry” AND “design”) AND (“ecology” OR “ecological” OR “ecological design” OR “regenerative” OR “eco-design” OR “ecofriendly”)	104

	(Sustainability and Design or Sustainable design) Or (Ecology and Design or Ecological design) in the context of Biomimicry	(“biomimicry” AND “design”) AND (“sustainability” OR “sustainable” OR “sustainable design”) OR (“ecology” OR “ecological” OR “eco-design” OR “ecofriendly” OR “regenerative” OR “ecological design”)	210
	(Sustainability and Design or Sustainable Design) And (Ecology and Design or Ecological Design) in the Context of Biomimicry	(“biomimicry” AND “design”) AND (“sustainability” OR “sustainable” OR “sustainable design”) AND (“ecology” OR “ecofriendly” or “ecological” OR “regenerative” OR “ecological design” OR “eco-design”)	49	

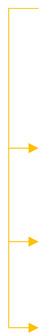	(Sustainability and Design or Sustainable design) Or (Ecology and Design or Ecological design) in Landscape Architecture in the context of Biomimicry	(“biomimicry” AND “design”) AND (“sustainability” OR “sustainable” OR “sustainable design”) OR (“ecology” OR “ecofriendly” or “ecological” OR “regenerative” OR “ecological design” OR “eco-design”) AND (“landscape architecture” OR “landscape design” OR “landscaping”)	3	
(Sustainability and Design or Sustainable design) in Landscape Architecture in the context of Biomimicry	(“biomimicry” AND “design”) AND (“sustainability” OR “sustainable” OR “sustainable design”) AND (“landscape architecture” OR “landscape design” OR “landscaping”)	1	
(Ecology and Design or Ecological design) in Landscape Architecture in the context of Biomimicry	(“biomimicry” AND “design”) AND (“ecology” OR “ecological” OR “ecological design” OR “eco-design” OR “regenerative” OR “ecofriendly”) AND (“landscape architecture” OR “landscape design” OR “landscaping”)	2	
(Sustainability and Design or Sustainable design) And (Ecology and Design or Ecological design) in Landscape Architecture in the context of Biomimicry	(“biomimicry” AND “design”) AND (“sustainability” OR “sustainable” OR “sustainable design”) AND (“ecology” OR “ecofriendly” or “ecological” OR “regenerative” OR “ecological design” OR “eco-design”) AND (“landscape architecture” OR “landscape design” OR “landscaping”)	0	

**Table 2 biomimetics-10-00559-t002:** Most frequently used words and their frequencies in the author keywords and titles of the 1634 documents.

Authors Keywords		Title	
Word	Frequency	Word	Frequency
bioinspiration	93	design	208
tissue engineering	70	bioinspiration	135
sustainability	69	engineering	101
biologically inspired design	47	applications	78
3d printing	37	tissue	73
design	30	materials	68
hydrogel	30	sustainability	68
biomaterials	29	review	67
self-assembly	26	nature	65
architecture	25	systems	51

## Data Availability

Data are contained within the article.

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
