# Peer review of "A Bibliometric Evaluation of the Use of Biomimicry as a Nature-Compatible Design Approach in Landscape Architecture Within the Context of Sustainability and Ecology"

_biomimetics, 2025, doi:10.3390/biomimetics10090559_

Round 1
Reviewer 1 Report (Previous Reviewer 2)
Comments and Suggestions for Authors
General Comments
In my opinion, the topic of the article is important and timely. The issue of biomimicry as an approach supporting design in landscape architecture can be a valuable response to contemporary civilizational challenges. The article attempts to quantitatively assess the presence of this approach in scientific literature, which may constitute a significant contribution to the further development of interdisciplinary research in these areas.
Specific Comments
- Topic.
The research area appears to be very interesting. The selection of the research focus is appropriate – the authors identify a research gap related to the lack of integration between biomimicry and landscape design in the literature, which is supported by the presented data. However, the topic does not fully correspond with the actual content. The title suggests an analysis combining biomimicry, design, landscape architecture, sustainability, and ecology. In reality, a significant portion of the results concern a bibliometric analysis of the broader field of biomimicry, and partially its connection with design, sustainability, and ecology. Very little space is devoted to the strictly defined research scope. In my view, this should be reflected either in the title of the article or clearly justified by the authors. This should also be explained in the introduction and methodology sections – why the authors analyze two article sets that may, but do not necessarily, include the intersection of biomimicry, design, landscape architecture, sustainability, and ecology, and what the purpose of exploring these areas is. Currently, Section 1.5 does not justify including the area of biomimicry in the argumentation. - Research Methodology.
There are inconsistencies in the description of the research methodology. For example, Figure 1 suggests that keywords were searched, while line 214 mentions a search in the article title, abstract, and keywords fields. - Research Results.
There are numerous issues with the readability of the graphics and visual data that describe the research results. For example (due to the frequency of such cases, the entire text should be reviewed thoroughly; I may not have caught them all): - The coding of article datasets in the results descriptions and the figures requires standardization. The term “210 documents” is used inconsistently in different parts of the article. The reader cannot be expected to infer which thematic area the publications belong to. The authors use different labels for the same dataset, which may hinder understanding, e.g., “data from 210 documents in which biomimicry, design, sustainability, and ecological design”; “Map of the countries' scientific productivity over time based on the analysis of 210 documents”; “Countries’ scientific productivity over time based on the data from 210 documents”; “In the subset of 210 documents”.
- In lines 360–361 and 511, it is unclear which documents are being referenced – the reader must deduce this from the numerical context.
- Figures 3, 4, 5, 8, 9, 10, 11, and 12 are difficult to read due to small font sizes, missing legends, or unclear data representation.
- Figure 2 does not indicate which research area the three publications combining all five elements belong to.
- Figure 7 lacks information on which group of documents the analysis refers to.
- Legends do not include references to relevant textual information (e.g., the note in line 387 should also appear in the legend for Figure 8).
- In Section 3.6.1, it is not specified which documents were the basis for the keyword co-occurrence analysis.
- Table 2 contains non-English words.
- Lack of content analysis.
The three publications that meet the criteria of biomimicry + design + landscape architecture + sustainability + ecology were not discussed in terms of their content or their relevance to the study's conclusions. While the article’s title implies that detailed content analysis will not be performed, given the very small number of publications strictly related to the study’s core, it would be worthwhile to include a few general parameters describing them.
The article addresses an important topic and offers a new perspective for biomimicry research. However, it requires further editorial, methodological, and structural revisions.
Author Response
Manuscript ID: biomimetics-3779176
Title: A Bibliometric Evaluation of Biomimicry as a Na-ture-Compatible Design in Landscape Architecture within the Context of Sustainability and Ecology
Dear Editor and reviewers
We would like to express our sincere gratitude to the editor and reviewers for their constructive comments and valuable suggestions. Their insightful feedback greatly contributed to the improvement of our manuscript. We have carefully revised the paper accordingly and addressed all comments in detail.
Responses to Reviewers
Reviewer 1
General Comments
In my opinion, the topic of the article is important and timely. The issue of biomimicry as an approach supporting design in landscape architecture can be a valuable response to contemporary civilizational challenges. The article attempts to quantitatively assess the presence of this approach in scientific literature, which may constitute a significant contribution to the further development of interdisciplinary research in these areas.
Specific Comments
- Topic.
The research area appears to be very interesting. The selection of the research focus is appropriate – the authors identify a research gap related to the lack of integration between biomimicry and landscape design in the literature, which is supported by the presented data. However, the topic does not fully correspond with the actual content. The title suggests an analysis combining biomimicry, design, landscape architecture, sustainability, and ecology. In reality, a significant portion of the results concern a bibliometric analysis of the broader field of biomimicry, and partially its connection with design, sustainability, and ecology. Very little space is devoted to the strictly defined research scope. In my view, this should be reflected either in the title of the article or clearly justified by the authors. This should also be explained in the introduction and methodology sections – why the authors analyze two article sets that may, but do not necessarily, include the intersection of biomimicry, design, landscape architecture, sustainability, and ecology, and what the purpose of exploring these areas is. Currently, Section 1.5 does not justify including the area of biomimicry in the argumentation.
- Author’s response: Thank you for your observation. We revised the title to better reflect the actual scope and content of the study. The revised title is:
"A Bibliometric Evaluation of The Use of Biomimicry as a Nature-Compatible Design Approach in Landscape Architecture within the Context of Sustainability and Ecology"
- This revised title aligns more closely with the bibliometric nature of the study and clarifies the inclusion of design, ecology, and sustainability. Additionally, clarifications were made in the Introduction and Section 1.5 to justify the two dataset strategy and the scope of inclusion.
- We revised Section 1.5 to include a clear rationale for selecting two article sets. The purpose of broadening the inclusion criteria is now explained as a way to trace the academic positioning of biomimicry within and beyond landscape architecture to identify gaps and trends. This addition improves transparency and methodological clarity.
- Research Methodology
There are inconsistencies in the description of the research methodology. For example, Figure 1 suggests that keywords were searched, while line 214 mentions a search in the article title, abstract, and keywords fields.
- Author’s response: the methodology section was revised to correct this inconsistency. Line 214 now correctly states that the search was conducted within "title, abstract, and keyword" fields in Scopus, and Figure 1 was updated accordingly to reflect this.
- Research Results
There are numerous issues with the readability of the graphics and visual data that describe the research results. For example (due to the frequency of such cases, the entire text should be reviewed thoroughly; I may not have caught them all):
- The coding of article datasets in the results descriptions and the figures requires standardization. The term “210 documents” is used inconsistently in different parts of the article. The reader cannot be expected to infer which thematic area the publications belong to. The authors use different labels for the same dataset, which may hinder understanding, e.g., “data from 210 documents in which biomimicry, design, sustainability, and ecological design”; “Map of the countries' scientific productivity over time based on the analysis of 210 documents”; “Countries’ scientific productivity over time based on the data from 210 documents”; “In the subset of 210 documents”.
- In lines 360–361 and 511, it is unclear which documents are being referenced – the reader must deduce this from the numerical context.
- Figures 3, 4, 5, 8, 9, 10, 11, and 12 are difficult to read due to small font sizes, missing legends, or unclear data representation.
- Figure 2 does not indicate which research area the three publications combining all five elements belong to.
- Figure 7 lacks information on which group of documents the analysis refers to.
- Legends do not include references to relevant textual information (e.g., the note in line 387 should also appear in the legend for Figure 8).
- In Section 3.6.1, it is not specified which documents were the basis for the keyword co-occurrence analysis.
- Table 2 contains non-English words.
- Author’s response :Thank you for your detailed feedback. We made the following corrections:
- Standardized terminology throughout the paper, consistently referring to the datasets (e.g., "210-document subset" or “Set 2”).
- Figures 3, 4, 5, 8–12 were updated with larger font sizes, clear legends, and consistent labeling.
- Figure 2 was annotated to clarify the scope of the 3 publications covering all five dimensions.
- Figure 7 now includes a clear note regarding which dataset it reflects.
- All legends were reviewed and improved to include connections to the main text.
- Section 3.6.1 now explicitly states which documents were used for co-occurrence analysis.
- Table 2 was revised to ensure all entries are in English.
- Lack of content analysis.
The three publications that meet the criteria of biomimicry + design + landscape architecture + sustainability + ecology were not discussed in terms of their content or their relevance to the study's conclusions. While the article’s title implies that detailed content analysis will not be performed, given the very small number of publications strictly related to the study’s core, it would be worthwhile to include a few general parameters describing them.
The article addresses an important topic and offers a new perspective for biomimicry research. However, it requires further editorial, methodological, and structural revisions.
Author’s response :In the Discussion section, we added a concise summary of the three studies that simultaneously include biomimicry, landscape architecture, sustainability, and ecology. These summaries provide a general overview of their approaches, contributions, and alignment with the objectives of our study.

Reviewer 2 Report (Previous Reviewer 1)
Comments and Suggestions for Authors
- Many non-English references have caused difficulties in reading this article. Please try to replace them with English literature as much as possible.
- Some non-English expressions in the text must also be translated into English.
- Reorganize the language, tables, figures, and the structure of the paper to enable readers to more easily and clearly understand your content and main findings.
- Line 192-193: Some of these databases or platforms are not comparable. Refer more to studies focusing on "Web of Science Core Collection" and Scopus published in established journals such as Scientometrics.
- Line 195-201: Some of your description of Scopus is not correct. Similarly, replace references 49-55 with more studies focusing on "Web of Science Core Collection" and Scopus published in established journals such as Scientometrics.
- Line 195-201: Some studies focusing on these databases also documented the widespread use of Web of Science and Scopus in academia. Disclose this point to readers and justify your choice of Scopus.
- Figure 4: The full names of the authors should be used. Be aware of the issue of name ambiguity.
- Line 342-345: The caveats of the affiliation country field in Scopus, as probed in studies such as "Red alert: Millions of “homeless” publications in Scopus should be resettled", should be disclosed to readers.
- The limitations of Scopus, as widely documented in the literature, such as "Web of Science and Scopus language coverage", should be added in the limitations section.
- Strengthen the dialogue with existing literature and highlight the new contributions of your research.
Author Response
Manuscript ID: biomimetics-3779176
Title: A Bibliometric Evaluation of Biomimicry as a Na-ture-Compatible Design in Landscape Architecture within the Context of Sustainability and Ecology
Dear Editor and reviewers
We would like to express our sincere gratitude to the editor and reviewers for their constructive comments and valuable suggestions. Their insightful feedback greatly contributed to the improvement of our manuscript. We have carefully revised the paper accordingly and addressed all comments in detail.
Responses to Reviewers
Reviewer 2
Comments and Suggestions for Authors
- Many non-English references have caused difficulties in reading this article. Please try to replace them with English literature as much as possible.
Author’s response :All non-English references were reviewed. Where possible, English equivalents from reputable sources (especially those indexed in Scopus/Web of Science) were substituted. A few non-English sources were retained only when equivalent English references could not be found.
- Some non-English expressions in the text must also be translated into English.
Author’s response : The manuscript was carefully revised to ensure all non-English phrases or terminologies were translated into English. This includes specific terminology in Table 2, figure notes, and section headings.
- Reorganize the language, tables, figures, and the structure of the paper to enable readers to more easily and clearly understand your content and main findings.
Author’s response : The entire manuscript was edited for clarity, consistency, and readability. The structure of tables and figures was harmonized with standard scientific formatting. Language was proofread for fluency and flow.
- Line 192-193: Some of these databases or platforms are not comparable. Refer more to studies focusing on "Web of Science Core Collection" and Scopus published in established journals such as Scientometrics.
- Line 195-201: Some of your description of Scopus is not correct. Similarly, replace references 49-55 with more studies focusing on "Web of Science Core Collection" and Scopus published in established journals such as Scientometrics.
- Line 195-201: Some studies focusing on these databases also documented the widespread use of Web of Science and Scopus in academia. Disclose this point to readers and justify your choice of Scopus.
Author’s response 4-6: In lines 192–201, we revised the paragraph to clearly distinguish between the functionalities of different databases. We also added citations from Scientometrics and Journal of Informetrics to justify our exclusive use of Scopus.
References 49–55 were updated with authoritative studies using Scopus and WoS in bibliometric research.
- Figure 4: The full names of the authors should be used. Be aware of the issue of name ambiguity.
Author’s response : Figure 4 was revised to include full author names, and name ambiguity issues were noted in the caption and explained briefly in the methodology.
- Line 342-345: The caveats of the affiliation country field in Scopus, as probed in studies such as "Red alert: Millions of “homeless” publications in Scopus should be resettled", should be disclosed to readers.
Author’s response : A relevant limitation was added to the Limitations section, noting the problem of inaccurate or missing country affiliations in Scopus, based on studies such as Red Alert: Millions of “Homeless” Publications in Scopus…
- The limitations of Scopus, as widely documented in the literature, such as "Web of Science and Scopus language coverage", should be added in the limitations section.
Author’s response : The Limitations section was expanded to include known constraints related to language coverage and journal indexing bias in Scopus. Citations were added to support this.
- Strengthen the dialogue with existing literature and highlight the new contributions of your research.
Author’s response: The Introduction, Discussion, and Conclusion sections were enriched with comparisons to similar bibliometric studies in design, sustainability, and ecology. The unique contribution of this study highlighting the minimal intersection of biomimicry with landscape design is now explicitly emphasized.

Round 2
Reviewer 1 Report (Previous Reviewer 2)
Comments and Suggestions for Authors
Thank you to the authors for implementing the revisions, which in my opinion enhance the scholarly value of the article—particularly the clarification of the title, the expanded description of the methodology, and the addition of content analysis of the three key publications. The authors address a timely and important research area: the assessment of the development and application of biomimicry in landscape design within a broader context, which contributes meaningfully to the literature. The article still presents certain weaknesses, such as the limited number of publications directly related to the full thematic scope, the incomplete standardization of terminology, and the low readability of some figures. Nevertheless, I believe the article also reflects the authors' individual approach to the problem and its interpretation, and that authors should be allowed a certain degree of freedom in these matters. Therefore, in my opinion, the article is suitable for publication.
Author Response
Dear Editor and reviewers
We would like to express our sincere gratitude to the editor and reviewers for their constructive comments and valuable suggestions, which have greatly contributed to improving our manuscript. We have carefully revised the paper accordingly and addressed all comments in detail. In the revised manuscript, all changes are highlighted in yellow in the Track Changes version. These highlighted sections indicate the parts that have been revised, corrected, or newly added in response to the reviewers’ feedback. Each change is explained in detail in the following point-by-point “Responses to Reviewers” section, with specific indications of where the revisions were made in the manuscript.
Responses to Reviewers
Reviewer 1
General Comments
In my opinion, the topic of the article is important and timely. The issue of biomimicry as an approach supporting design in landscape architecture can be a valuable response to contemporary civilizational challenges. The article attempts to quantitatively assess the presence of this approach in scientific literature, which may constitute a significant contribution to the further development of interdisciplinary research in these areas.
Specific Comments
- Topic.
The research area appears to be very interesting. The selection of the research focus is appropriate – the authors identify a research gap related to the lack of integration between biomimicry and landscape design in the literature, which is supported by the presented data. However, the topic does not fully correspond with the actual content. The title suggests an analysis combining biomimicry, design, landscape architecture, sustainability, and ecology. In reality, a significant portion of the results concern a bibliometric analysis of the broader field of biomimicry, and partially its connection with design, sustainability, and ecology. Very little space is devoted to the strictly defined research scope. In my view, this should be reflected either in the title of the article or clearly justified by the authors. This should also be explained in the introduction and methodology sections – why the authors analyze two article sets that may, but do not necessarily, include the intersection of biomimicry, design, landscape architecture, sustainability, and ecology, and what the purpose of exploring these areas is. Currently, Section 1.5 does not justify including the area of biomimicry in the argumentation.
- Author’s response: Thank you for your observation. We revised the title to better reflect the actual scope and content of the study. The revised title is:
"A Bibliometric Evaluation of The Use of Biomimicry as a Nature-Compatible Design Approach in Landscape Architecture within the Context of Sustainability and Ecology"
- This revised title aligns more closely with the bibliometric nature of the study and clarifies the inclusion of design, ecology, and sustainability. Additionally, clarifications were made in the Introduction and Section 1.5 to justify the two dataset strategy and the scope of inclusion.
- We revised Section 1.5 to include a clear rationale for selecting two article sets. The purpose of broadening the inclusion criteria is now explained as a way to trace the academic positioning of biomimicry within and beyond landscape architecture to identify gaps and trends. This addition improves transparency and methodological clarity.
- Research Methodology
There are inconsistencies in the description of the research methodology. For example, Figure 1 suggests that keywords were searched, while line 214 mentions a search in the article title, abstract, and keywords fields.
- Author’s response: the methodology section was revised to correct this inconsistency. Line 214 now correctly states that the search was conducted within "title, abstract, and keyword" fields in Scopus, and Figure 1 was updated accordingly to reflect this.
- Research Results
There are numerous issues with the readability of the graphics and visual data that describe the research results. For example (due to the frequency of such cases, the entire text should be reviewed thoroughly; I may not have caught them all):
- The coding of article datasets in the results descriptions and the figures requires standardization. The term “210 documents” is used inconsistently in different parts of the article. The reader cannot be expected to infer which thematic area the publications belong to. The authors use different labels for the same dataset, which may hinder understanding, e.g., “data from 210 documents in which biomimicry, design, sustainability, and ecological design”; “Map of the countries' scientific productivity over time based on the analysis of 210 documents”; “Countries’ scientific productivity over time based on the data from 210 documents”; “In the subset of 210 documents”.
- In lines 360–361 and 511, it is unclear which documents are being referenced – the reader must deduce this from the numerical context.
- Figures 3, 4, 5, 8, 9, 10, 11, and 12 are difficult to read due to small font sizes, missing legends, or unclear data representation.
- Figure 2 does not indicate which research area the three publications combining all five elements belong to.
- Figure 7 lacks information on which group of documents the analysis refers to.
- Legends do not include references to relevant textual information (e.g., the note in line 387 should also appear in the legend for Figure 8).
- In Section 3.6.1, it is not specified which documents were the basis for the keyword co-occurrence analysis.
- Table 2 contains non-English words.
- Author’s response :Thank you for your detailed feedback. We made the following corrections:
- Standardized terminology throughout the paper, consistently referring to the datasets (e.g., "210-document subset" or “Set 2”).
- Figures 3, 4, 5, 8–12 were updated with larger font sizes, clear legends, and consistent labeling.
- Figure 2 was annotated to clarify the scope of the 3 publications covering all five dimensions.
- Figure 7 now includes a clear note regarding which dataset it reflects.
- All legends were reviewed and improved to include connections to the main text.
- Section 3.6.1 now explicitly states which documents were used for co-occurrence analysis.
- Table 2 was revised to ensure all entries are in English.
- Lack of content analysis.
The three publications that meet the criteria of biomimicry + design + landscape architecture + sustainability + ecology were not discussed in terms of their content or their relevance to the study's conclusions. While the article’s title implies that detailed content analysis will not be performed, given the very small number of publications strictly related to the study’s core, it would be worthwhile to include a few general parameters describing them.
The article addresses an important topic and offers a new perspective for biomimicry research. However, it requires further editorial, methodological, and structural revisions.
Author’s response :In the Discussion section, we added a concise summary of the three studies that simultaneously include biomimicry, landscape architecture, sustainability, and ecology. These summaries provide a general overview of their approaches, contributions, and alignment with the objectives of our study.

Reviewer 2 Report (Previous Reviewer 1)
Comments and Suggestions for Authors
- Many of my suggestions for the previous round of review have not been addressed. These suggestions were merely mentioned as having been adopted in the response letter, but were not adopted in the revised manuscript.
- The limitations of the Scopus database, as mentioned in the first round of review, should be added. For example, the caveats of the affiliation country field in Scopus and the regional bias of Scopus have been probed by many previous studies. Many related studies focusing on Scopus can be found in established journals such as Scientometrics, Journal of the Association for Information Science and Technology, and Journal of Informetrics.
- Many figures are not clear to read.
- Please refer to some classic bibliometric studies and optimize the structure of the article.
- Many references should be replaced with authoritative ones. Try to avoid using non-English literature, unless it is necessary.
- It is hoped that the authors will take the suggestions into consideration or provide a reasonable explanation.
Author Response
We would like to express our sincere gratitude to the editor and reviewers for their constructive comments and valuable suggestions, which have greatly contributed to improving our manuscript. We have carefully revised the paper accordingly and addressed all comments in detail. In the revised manuscript, all changes are highlighted in yellow in the Track Changes version. These highlighted sections indicate the parts that have been revised, corrected, or newly added in response to the reviewers’ feedback. Each change is explained in detail in the following point-by-point “Responses to Reviewers” section, with specific indications of where the revisions were made in the manuscript.
Responses to Reviewers
Reviewer 2
Comments and Suggestions for Authors
- Many non-English references have caused difficulties in reading this article. Please try to replace them with English literature as much as possible.
Author’s response :All non-English references were reviewed. Where possible, English equivalents from reputable sources (especially those indexed in Scopus/Web of Science) were substituted. A few non-English sources were retained only when equivalent English references could not be found.
- Some non-English expressions in the text must also be translated into English.
Author’s response : The manuscript was carefully revised to ensure all non-English phrases or terminologies were translated into English. This includes specific terminology in Table 2, figure notes, and section headings.
- Reorganize the language, tables, figures, and the structure of the paper to enable readers to more easily and clearly understand your content and main findings.
Author’s response : The entire manuscript was edited for clarity, consistency, and readability. The structure of tables and figures was harmonized with standard scientific formatting. Language was proofread for fluency and flow.
- Line 192-193: Some of these databases or platforms are not comparable. Refer more to studies focusing on "Web of Science Core Collection" and Scopus published in established journals such as Scientometrics.
- Line 195-201: Some of your description of Scopus is not correct. Similarly, replace references 49-55 with more studies focusing on "Web of Science Core Collection" and Scopus published in established journals such as Scientometrics.
- Line 195-201: Some studies focusing on these databases also documented the widespread use of Web of Science and Scopus in academia. Disclose this point to readers and justify your choice of Scopus.
Author’s response 4-6: In lines 192–201, we revised the paragraph to clearly distinguish between the functionalities of different databases. We also added citations from Scientometrics and Journal of Informetrics to justify our exclusive use of Scopus.
References 49–55 were updated with authoritative studies using Scopus and WoS in bibliometric research.
- Figure 4: The full names of the authors should be used. Be aware of the issue of name ambiguity.
Author’s response : Figure 4 was revised to include full author names, and name ambiguity issues were noted in the caption and explained briefly in the methodology.
- Line 342-345: The caveats of the affiliation country field in Scopus, as probed in studies such as "Red alert: Millions of “homeless” publications in Scopus should be resettled", should be disclosed to readers.
Author’s response : A relevant limitation was added to the Limitations section, noting the problem of inaccurate or missing country affiliations in Scopus, based on studies such as Red Alert: Millions of “Homeless” Publications in Scopus…
- The limitations of Scopus, as widely documented in the literature, such as "Web of Science and Scopus language coverage", should be added in the limitations section.
Author’s response : The Limitations section was expanded to include known constraints related to language coverage and journal indexing bias in Scopus. Citations were added to support this.
- Strengthen the dialogue with existing literature and highlight the new contributions of your research.
Author’s response: The Introduction, Discussion, and Conclusion sections were enriched with comparisons to similar bibliometric studies in design, sustainability, and ecology. The unique contribution of this study—highlighting the minimal intersection of biomimicry with landscape design—is now explicitly emphasized.

This manuscript is a resubmission of an earlier submission. The following is a list of the peer review reports and author responses from that submission.
Round 1
Reviewer 1 Report
Comments and Suggestions for Authors
This study conducted a stepwise bibliometric analysis using the Scopus database to quantitatively explore the relationship between biomimicry and sustainable/ecological design in landscape architecture. A few suggestions:
- I suggest that the authors refer to more classic studies in bibliometrics and conduct more rigorous bibliometric research, including the accuracy of language usage, the selection of indicators, and the optimization of structure. Some classical studies published in Scientometrics are helpful.
- Figure 1 seems to be incomplete.
- Table 1 is difficult to understand.
- The structure of the article and the titles of the paragraphs need to be optimized.
- The features and limitations of the Scopus database need to be disclosed in detail. Some studies from Journal of the Association for Information Science and Technology, Scientometrics, and Journal of Informetrics may be helpful.
Reviewer 2 Report
Comments and Suggestions for Authors
The subject of the article is very interesting and has developmental potential. It addresses an important and contemporary issue: the integration of the concepts of biomimicry, sustainable/ecological design, and landscape architecture. The authors clearly identify a research gap and present a sound, though methodologically limited, approach (they do not apply more advanced and commonly used bibliometric methods based on network analysis, such as co-occurrence and keyword clustering, co-citation analysis, or co-authorship mapping). The methodology is based on data from the Scopus database and bibliometric tools available in the Bibliometrix package.
The visualizations presented in the article are clear, and the interpretation of the data is logical and systematic. However, the main limitation of the work—which significantly affects its scope and impact—is the very low number of publications (only 3–4) that meet the core thematic criteria defined by the authors. This raises a key question: are we dealing with a true and significant research gap, or rather with limitations in the applicability of biomimicry in landscape design practice, or even with terminological and methodological challenges in identifying relevant literature?
The authors adopt the interpretation that a research gap exists, but they do not provide sufficiently strong arguments to support the potential of biomimicry in landscape design. Their reasoning is too general and declarative and is not supported by evidence or sufficient discussion.
In my opinion, the authors should further develop the discussion on this point and indicate specific reasons why biomimicry should be developed in the context of landscape design. A more in-depth discussion would be advisable—ideally including references to potential applications, design strategies, or environmental challenges that could be addressed through biomimetic approaches.
I would like to thank the authors for their work in preparing this article and see this manuscript published but considering at least the above suggestion.